# Risk factors for the progression to multimorbidity among UK urban working-age adults. A community cohort study

Anne L. Stagg [1,2]*, Lisa Harber-Aschan[1,3], Stephani L. Hatch[1], Nicola T. Fear[4,5], Sarah Dorrington[1,6], Ira Madan[2], Sharon A. M. Stevelink[1,5]

**1** Department of Psychological Medicine, Institute of Psychiatry Psychology & Neuroscience, King's College London, London, United Kingdom, **2** Department of Occupational Health, Guy's and St Thomas' Hospitals NHS Foundation Trust, London, United Kingdom, **3** Stockholm University Demography Unit, Stockholm University, Stockholm, Sweden, **4** Academic Department of Military Mental Health, King's College London, London, United Kingdom, **5** King's Centre for Military Health Research, King's College London, London, United Kingdom, **6** Biomedical Research Centre, South London and Maudsley NHS Foundation Trust, London, United Kingdom

* anne.l.stagg@kcl.ac.uk

**Data Availability Statement:** This study used third party data from the SELCoH survey. SELCoH data is available to researchers upon request. Please

## Abstract

### Objectives

The progression of long-term conditions (LTCs) from zero-to-one (initiation), and from one-to-many (progression) are common trajectories that impact a person's quality of life including their ability to work. This study aimed to explore the demographic, socioeconomic, psycho-social, and health-related determinants of LTC initiation and progression, with a focus on work participation.

### Methods

Data from 622 working-age adults who had completed two waves (baseline and follow-up) of the South-East London Community Health survey were analysed. Chi square tests and multinomial logistic regression were used to describe the associations between self-reported demographic, socioeconomic, psychosocial, and health-related variables, and the progression of LTCs.

### Results

Small social networks, an increased number of stressful life events, low self-rated health, functional impairment, and increased somatic symptom severity were all associated with both the progression from zero-to-one LTC and from one LTC to multimorbidity (two or more LTCs). Renting accommodation (RRR 1.73 [95% CI 1.03–2.90]), smoking (RRR 1.91 [95% CI 1.16–3.14]) and being overweight (RRR 1.88 [95% CL 1.12–3.16]) were unique risk factors of developing initial LTCs, whereas low income (RRR 2.53 [95% CI 1.11–5.80]), working part-time (RRR 2.82 ([95% CL 1.12–7.10]), being unemployed (RRR 4.83 [95% CI 1.69–13.84]), and making an early work exit (RRR 16.86 [95% CI 3.99–71.30]) all increased the risk of progressing from one LTC to multimorbidity compared to being employed full-time. At

contact selcoh@kcl.ac.uk to apply for access. The authors confirm they did not receive any special privileges in accessing the data that other researchers would not have. For more information visit www.kcl.ac.uk/research/selcoh.

**Funding:** This study was funded by Guy's and St Thomas' Charity, EIC180702: IM and SAMS. SAMS is supported by the National Institute for Health Research (NIHR) Maudsley Biomedical Research Centre at South London and Maudsley NHS Foundation Trust and the National Institute for Health Research, NIHR Advanced Fellowship, Dr Sharon A M Stevelink, NIHR300592. SH is supported by the ESRC Centre for Society and Mental Health at King's College London (ESRC Reference: ES/S012567/1) and the NIHR Biomedical Research Centre at South London and Maudsley NHS Foundation Trust. The funders had no role in study design, data collection and analysis, decision to publish, or preparation of the manuscript.

**Competing interests:** This paper represents independent research part funded by NIHR Maudsley Biomedical Research Centre at South London and Maudsley NHS Foundation Trust and King's College London, and the NIHR. The views expressed are those of the authors and not necessarily those of the NHS, the NIHR or the Department of Health and Social Care. This does not alter our adherence to PLOS ONE policies on sharing data and materials.

follow-up, depression was the most prevalent LTC in the unemployed group whereas musculoskeletal conditions were the most prevalent in those working.

## Conclusions

The journey to multimorbidity is complex, with both common and unique risk factors. Non-full-time employment was associated with an increased risk of progression to multimorbidity. Future research should explore the risk and benefit pathways between employment and progression of LTCs. Interventions to prevent progression of LTCs should include mitigation of modifiable risk factors such as social isolation.

## Introduction

The number of people living with multiple long-term health conditions (MLTCs, multimorbidity) is rising, with older age, socioeconomic deprivation and poor health behaviours being key risk factors [1–4]. The journey to deteriorating health often starts with the acquisition of an initial long-term condition (LTC), with additional LTCs developing over time (negative health transitions), leading to an individual living with the impact of multimorbidity [5]. Multimorbidity is now developing earlier in the life course; studies in England and Scotland have shown that in absolute numbers, there are more people with multimorbidity below the age of 65 than over 65 [2, 6]. Although the development of LTCs increases with age, the current research focus on LTCs in the older age groups does not explore the risk factors for LTC acquisition or the impact of living with MLTCs in younger, working-age adults. Indeed, it has been reported that more than a third of patients with three-or-more LTCs that lead to severe functional impairment and high health and social care requirements, are between the ages of 18 and 64 years [5].

There is a close relationship between quality of life and functional status, with one key indicator of functional status being the ability to engage in employment. The association between employment and long-term conditions is complex and known to have a bi-directional relationship [7]. Health conditions can impact workers' ability to engage in employment, but unemployment and poor working conditions among those in work, can also impact health [8, 9]. Unemployment is associated with poorer physical and mental health, with the risk of having both one and multiple LTCs greater in those who are unemployed compared to employed [8, 10]. However, the impact of employment on the development of initial and multiple LTCs is under-researched, and few studies focus exclusively on working-age adults [8].

Among working-age adults, known risk factors for multimorbidity not only include increasing age and socioeconomic deprivation, but also obesity and smoking [6, 11]. However, whilst there is evidence that socioeconomic deprivation influences the transition to multimorbidity [4], there is a lack of information about individual socioeconomic, psychosocial, behavioural, and health-related influencers of developing a first LTC, and of progression from one-to-many LTCs. The objective of the present study is to address this research gap by exploring a wide range of potential risk factors and LTC development and progression within a working-age community sample. By using two time points, the primary aim of this study is to explore relationships between demographic, socioeconomic (including employment status), psychosocial (social isolation and stressful life events), and health-related determinants at the baseline, and two negative health transitions between baseline and follow-up of (1) initiation (0 to one-or-more LTCs), and (2) progression of LTCs (one LTC to multimorbidity). This objective also aims to gain a better understanding of demographic and socioeconomic inequalities that

influence health disparities in the development and progression of LTCs [12]. The secondary aim is to provide information about the distribution of the most common LTCs at follow-up within the different categories of employment status.

## Methods

### Study design, setting and participants

This paper utilises data from adults aged 16 to 64 years who participated in the South East London Community Health (SELCoH) study [13, 14]. The SELCoH study assessed physical and psychiatric morbidity in adults aged 16 and overliving in randomly selected households in Lambeth and Southwark. Survey data from the first (baseline) and second waves (follow-up) of the SELCoH study (S1 and S2) were used in which household interviews were carried out in 2008–2010 and 2011–2013. Demographic characteristics of participants who responded at both waves, and reasons for non-response at the second wave have previously been reported [14, 15]. Of the 1052 adults who completed both surveys, the present study used data from working-age adults who completed face-to-face interviews at both baseline and follow-up, and were within the age range of 16 years at baseline and 64 years at follow-up.

Ethical approval for the baseline SELCoH survey was given by King's College London Research Ethics Committee (CREC/07/08-152), and the follow-up survey wave by King's College London Psychiatry Nursing and Midwifery Research Ethics Committee (PNM/10/11-106). Participants gave written informed consent to participate in the study before taking part.

### Study variables

**Long-term health conditions.** In the SELCoH questionnaire, a long-term condition was defined as a long-standing illness, disability or infirmity troubling a participant over a period of time, or considered likely to in the future. Participants were able to select a specific LTC from a list of 15 health conditions, and in addition, specify any other condition not on the list. The final list of LTCs was constructed by allocating conditions either into specific condition groups (for example, two condition groups were asthma and high blood pressure), or into groups of similar conditions where individual case numbers were <5 (for example, gastrointestinal conditions and respiratory conditions). No participant had more than one condition within any of the individual combined condition groups. A final list of 17 conditions/ groups of conditions was derived from all self-reported conditions in addition to a general non-specific LTC group for conditions with less than 5 cases that could not be combined with other similar conditions. A summary description of the LTC groups has been previously published [10].

**Independent variables.** All variables were measured at the baseline (S1) except for ethnicity which was assessed at the follow-up interviews (S2). Demographic variables consisted of gender (male; female), age, relationship status, ethnicity, and migrant status. Due to small counts for LTCs in younger participants, age was categorised into four groups: 16–34, 35–44, 45–54 and 55–64 years. Relationship status was categorised into three groups comprising married or cohabiting, single and never married, and divorced, separated, or widowed. Ethnicity was categorised into three groups comprising White British, Black African / Black Caribbean, and Other ethnic groups. Migrant status was determined using country of birth and number of years residing in the UK and was categorised into 3 groups: non-migrants (born in the UK), migrants living in the UK for ≤ 10 years, and migrants living in the UK for ≥ 11 years.

Socioeconomic variables included housing tenure, highest education level attained, gross annual household income, employment, and benefit receipt. Housing tenure was reported as

owning a property with or without a mortgage, paying rent or part rent, and rent-free accommodation. Educational attainment was categorised into 4 categories: degree level or above, A level or equivalent, GCSE level or equivalent, and no qualifications. Gross annual household income was based on four income bands: (1) £31,495 or more (2) £12,098-£31,494 (3) £0–12,097 and (4) Don't know. Employment was categorised into working full time, working part time or casual work, unemployed, students, and two economically inactive groups: early work exit and 'at home'. Early work exit included those who were not available for work due to either having chosen to retire early (before age 64) or could not work due to being registered disabled or permanently sick. 'At home' indicated those who were at home looking after children. Benefit receipt was categorised into a binary variable of benefit receipt/no benefit receipt and excluded benefits relating to being sick or disabled due to potential increased predictor-outcome associations between benefit receipt and long-term condition status.

Psychosocial variables included social network size and stressful life events. Social network size was determined from the total number of weekly contacts with different groups of people, and categorised into three groups: $\geq$ 5, 3–4, and $\leq$ 2. The number of stressful life events was assessed using a checklist of childhood and lifetime events considered relevant to urban populations [16, 17], and the cumulative score categorised into $\leq$ 2, 2–5 and $\geq$ 6.

Health risk factors were body mass index (BMI), smoking and alcohol intake. BMI was calculated from weight and height measurements taken at the initial interview. Categorisation of BMI into four groups were: underweight ($<$18.4 kg/m$^2$), normal weight (18.5–24.9 kg/m$^2$), overweight (25.0–29.9 kg/m$^2$), and obese ($\geq$30.0 kg/m$^2$). Smoking was categorised as non-smokers (including past smokers), and current smokers (regular and sporadic). Hazardous alcohol intake was determined from a cumulative score of 8–40 on the Alcohol Use Disorders Identification Test (AUDIT) questionnaire, and moderate intake indicated by a score of 1–7 [13, 18, 19].

Health-related quality of life was measured using three individual items on the 12-item Short Form (SF-12) questionnaire to assess self-rated health, and subjective measures of functional limitations affecting the ability to work or other activities due to physical and emotional health (anxiety or depression) during the previous four weeks [20]. The 5-point measure of self-rated health was categorised into three groups: excellent/very good, good, and fair/poor, whereas limitations due to physical and emotional health were reported as binary responses of no/yes.

Somatic symptom severity was measured using the cumulative score from 14 of the 15 items in the Patient Health Questionnaire (PHQ-15, [21], excluding the question relating to menstrual cramps due to only being applicable to females [15]. Consistent with a previous study using the SELCoH community population, the cumulative score was categorised into low ($<$5), medium (5–9) and high (10–28) somatic symptom severity [15]).

**Progression of LTCs.** The number of LTCs were categorised at both baseline and follow-up to enable the investigation of negative health transitions. This comprised a baseline comparable 'healthy' group with no LTCs at baseline or follow-up, and two groups that showed negative health transitions; a group who progressed from no LTCs at baseline to one-or-more LTCs at follow-up (initiation group), and a group who progressed from one LTC at baseline to two-or-more LTCs at follow-up (progression group). Data from those who had more than one LTCs at baseline, or who had one-or-more LTCs at baseline but either remained stable or reported fewer or no LTCs at follow-up (n = 271), were excluded from the statistical analysis since the focus of this study was on factors associated with negative health transitions. The final LTC trajectory sample was 622 and the percentage of missing data was below 1.5%.

## Statistical analyses

All analyses were carried out using Stata/MP 16.1, applying robust standard errors using survey commands (svy) for prevalence estimates and associations. Weighting was applied to adjust for household clustering, non-response, attrition, and changes in household composition between baseline and follow-up. Descriptive statistics are reported using unweighted frequencies and weighted percentages, and Pearson's $\chi^2$ tests with Rao & Scott corrections for survey data. Multinomial logistic regressions were used to analyse associations between each individual indicator variable measured at baseline with the three categories of (1) no LTC at either baseline or follow-up (the reference category), (2) development from no LTCs at baseline to 1-or-more LTCs at follow-up (initiation) and (3) progression from 1 LTC at baseline and multiple LTCs at follow-up. Results from multinomial logistic regression report relative risk ratios (RRR) and 95% confidence intervals. for LTC initiation and progression between baseline and follow-up (S1 and S2). Two models were tested; unadjusted and also adjusted for possible confounding by gender and age (as a continuous variable, in years). The overall distribution of the eight most prevalent LTCs within each employment category at follow-up are reported and compared descriptively.

## Results

### LTCs and health transitions

In the whole sample (n = 893), the healthy group of participants with no LTCs at either baseline or follow-up accounted for 56% (n = 479), whereas 15.4% (n = 143) of the sample reported negative health transitions with 10.5% (n = 97) developing LTCs at follow-up, and 4.9% (n = 46) progressed from one-to-many LTCs between baseline and follow-up. Positive health transitions were also reported, with 52 participants (5.9%) reporting at least one LTC at baseline but no LTCs at follow-up, and 31 participants (3.0%) reporting two or more LTCs at baseline had an improved LTC status at follow-up, only reporting one LTC. 77 participants reported having no LTCs at baseline but a first LTC at follow-up, with depression (*n* = 15, 21.5%), musculoskeletal conditions (*n* = 13, 14.3%), asthma (*n* = 10, 12.8%) and high blood pressure (*n* = 8, 9.7%) being the most prevalent first conditions to be reported. The six most prevalent conditions in those reporting progressing from one to MLTCs at follow-up were musculoskeletal conditions (n = 21, 42.6%), depression (n = 16, 36.5%), asthma (n = 10, 22.6%), gastrointestinal conditions (n = 10, 24.7%), high blood pressure (n = 10, 20.8%) and migraine (n = 10, 22.1%).

The distribution of the stable and negative LTC trajectories used in the analysis are shown in Table 1 (*n* = 622). For these three LTC trajectories, the average time between S1 and S2 was 2.46 years (SD ± 0.55).

### Risk factors of negative health transitions

The unadjusted and adjusted relative risk ratios and 95% confidence intervals are shown in Table 2, with values adjusted for age and gender. Age was the primary demographic indicator of negative health transitions, with the 45–54 and 55–64 age group having an increased the risk of developing initial LTCs between baseline and follow-up. All age groups were at increased risk of progressing from one-to-many LTCs when compared to the youngest age band of 16–34 years. Once age and gender were accounted for in the adjusted model, being single was also associated with an increased risk of progressing from one-to-many LTCs.

Psychosocial factors increased the risk of both negative health transitions. Having the smallest social network ($\leq$ 2 known contacts per week) and being in the group with the greatest

**Table 1. Baseline demographic, socioeconomic, employment, psychosocial and health-related factor distribution of LTC trajectories.**

| | No LTCs [a] | 0-to- $\geq$ 1 LTC [b] | 1-to-MLTCs [c] | |
|---|---|---|---|---|
| | (n = 479) | (n = 97) | (n = 46) | |
| | *n* (row %) | *n* (row%) | *n* (row %) | $\chi^2$ *p-value* [d] |
| **Demographic variables** | | | | |
| Gender | | | | 0.390 |
| Female | 269 (76.2) | 62 (16.4) | 28 (7.4) | |
| Male | 210 (80.8) | 35 (12.9) | 18 (6.3) | |
| Age (years) | | | | **<0.001** |
| 16–34 | 284 (85.1) | 38 (11.5) | 11 (3.5) | |
| 35–44 | 106 (76.0) | 22 (15.7) | 11 (8.3) | |
| 45–54 | 68 (61.9) | 26 (23.0) | 16 (15.1) | |
| 55–64 | 21 (53.1) | 11 (26.5) | 8 (20.4) | |
| Marital status | | | | 0.072 |
| Married/Cohabiting | 234 (77.5) | 53 (16.1) | 21 (6.4) | |
| Single | 209 (81.3) | 33 (12.5) | 17 (6.2) | |
| Divorced/Separated/Widowed | 36 (65.1) | 11 (20.0) | 8 (15.0) | |
| Ethnicity | | | | 0.180 |
| White British | 229 (78.5) | 44 (14.1) | 23 (7.4) | |
| Black Caribbean & Black African | 119 (84.1) | 20 (12.5) | 5 (3.4) | |
| Other | 131 (73.7) | 33 (17.4) | 18 (8.9) | |
| Migrant Status | | | | 0.066 |
| Born in UK | 303 (77.8) | 62 (14.8) | 32 (7.4) | |
| 0–10 years in the UK | 107 (85.9) | 15 (11.2) | 4 (2.9) | |
| 11 or more years in the UK | 68 (70.5) | 20 (19.4) | 10 (10.2) | |
| **Socioeconomic variables** | | | | |
| Household income (gross, annual) | | | | 0.169 |
| £31,495 or more | 261 (80.8) | 47 (13.6) | 20 (5.6) | |
| £12,098–31,494 | 93 (75.3) | 28 (20.0) | 7 (4.7) | |
| £0–12,097 | 68 (75.1) | 12 (12.8) | 11 (12.2) | |
| Do not know | 56 (77.7) | 10 (13.1) | 8 (9.2) | |
| Housing tenure | | | | |
| Own/ mortgage | 164 (78.5) | 30 (12.4) | 20 (9.2) | 0.110 |
| Rent/part rent | 265 (76.5) | 62 (17.0) | 25 (6.6) | |
| Rent free | 49 (88.9) | 5 (9.3) | 1 (1.9) | |
| Education (highest level attained) | | | | 0.231 |
| Degree level or above | 249 (81.9) | 40 (12.5) | 19 (5.6) | |
| Up to A level | 123 (77.1) | 29 (16.1) | 13 (6.9) | |
| Up to GCSE | 78 (76.3) | 17 (14.8) | 9 (8.9) | |
| No qualifications | 27 (64.0) | 11 (24.6) | 5 (11.4) | |
| Employment | | | | **<0.001** |
| Employed full-time | 235 (81.5) | 46 (15.1) | 11 (3.3) | |
| Employed part-time | 83 (79.8) | 19 (15.9) | 12 (9.3) | |
| Unemployed | 39 (67.5) | 13 (20.5) | 7 (12.1) | |
| Student | 97 (87.9) | 8 (7.2) | 6 (4.9) | |
| Early work exit[e] | 5 (31.8) | 4 (24.1) | 7 (44.0) | |
| At home | 20 (67.4) | 7 (23.4) | 3 (9.2) | |
| Benefit receipt (not health-related) | | | | **0.023** |
| No benefits | 400 (80.5) | 72 (13.5) | 34 (5.9) | |
| Benefits | 79 (69.1) | 25 (19.8) | 12 (11.1) | |

*(Continued)*

**Table 1.** (Continued)

| | No LTCs [a] (n = 479) | 0-to- $\geq$ 1 LTC [b] (n = 97) | 1-to-MLTCs [c] (n = 46) | |
|---|---|---|---|---|
| | n (row %) | n (row%) | n (row %) | $\chi^2$ p-value [d] |
| **Psychosocial variables** | | | | |
| Size of social network | | | | **0.002** |
| $\geq$5 | 358 (81.3) | 62 (13.0) | 29 (5.8) | |
| 3–4 | 104 (76.1) | 25 (16.7) | 11 (7.2) | |
| $\leq$2 | 17 (51.6) | 10 (28.5) | 6 (20.0) | |
| Stressful life events | | | | **0.003** |
| $\leq$2 | 202 (83.8) | 32 (12.6) | 9 (3.6) | |
| 3–5 | 207 (78.5) | 40 (14.2) | 22 (7.4) | |
| $\geq$6 | 70 (66.1) | 25 (20.8) | 15 (13.0) | |
| **Health Risk Factors** | | | | |
| Body Mass Index | | | | **0.031** |
| Normal weight | 254 (81.7) | 35 (10.8) | 25 (7.6) | |
| Underweight | 14 (100) | 0 (0) | 0 (0) | |
| Overweight | 140 (73.3) | 41 (19.9) | 14 (6.8) | |
| Obese | 63 (73.7) | 19 (19.8) | 7 (7.0) | |
| Smoking | | | | |
| Non-smoker | 376 (80.9) | 66 (12.9) | 32 (6.2) | **0.033** |
| Current Smoker | 103 (70.8) | 31 (20.2) | 14 (8.9) | |
| Alcohol Use | | | | 0.708 |
| Non-drinkers (AUDIT score 0) | 91 (81.4) | 15 (12.7) | 7 (5.9) | |
| Moderate alcohol use (1–7) | 289 (78.8) | 62 (15.0) | 26 (6.2) | |
| Hazardous alcohol use (8–40) | 99 (75.2) | 20 (15.4) | 13 (9.4) | |
| **Health-Related Quality of Life** | | | | |
| Self-rated health | | | | **<0.001** |
| Excellent/very good | 299 (86.9) | 39 (10.6) | 10 (2.6) | |
| Good | 153 (72.6) | 43 (18.5) | 21 (8.9) | |
| Fair/Poor | 27 (50.4) | 15 (24.7) | 15 (25.0) | |
| Functional Impairment due to physical health | | | | **<0.001** |
| No | 446 (80.6) | 84 (14.0) | 34 (5.4) | |
| Yes | 33 (56.6) | 13 (21.8) | 12 (21.6) | |
| Functional impairment due to emotional health | | | | **0.002** |
| No | 429 (80.9) | 74 (13.0) | 37 (6.1) | |
| Yes | 50 (62.7) | 23 (25.3) | 9 (12.1) | |
| **Somatic symptoms** | | | | |
| Somatic symptom severity | | | | **<0.001** |
| Low (0/4) | 347 (83.9) | 54 (12.1) | 19 (4.1) | |
| Medium (5/9) | 107 (70.3) | 29 (17.2) | 20 (12.5) | |
| High ($\geq$10) | 25 (56.6) | 14 (29.6) | 7 (13.8) | |

Frequencies are unweighted and weighted percentages account for survey design. Numbers may not add up to the total sample (N = 622) due to missing data. All variables were measured at baseline (S1) except ethnicity which was determined at follow-up (S2).

[a] No LTCs: S1 = 0, S2 = 0

[b] 0-to- $\geq$ 1 LTC: S1 = 0 LTC, S2 $\geq$1 LTCs

[c] 1-to-MLTCs: S1 = 1 LTC, S2 $\geq$2 LTCs

[d] Pearson's $\chi 2$ test with Rao & Scott correction for survey data. Bold indicates significance ($p < 0.05$)

[e] Early work exit: permanently sick/disabled or early retirement

**Table 2.** Unadjusted and adjusted[a] multinomial regression analysis estimating associations between demographic, socioeconomic, psychosocial and health factors at baseline, and LTC trajectory groups.

| | Unadjusted | | Adjusted | |
| --- | --- | --- | --- | --- |
| | 0 to ≥1 LTCs[b] | 1-to-Many LTCs[c] | 0 to ≥1 LTCs[b] | 1-to-Many LTCs[c] |
| | RRR (95% CI) | RRR (95% CI) | RRR (95% CI) | RRR (95% CI) |
| **Demographic variables** | | | | |
| Gender (*n* = 622) | | | | |
| Female | 1.00 | 1.00 | 1.00 | 1.00 |
| Male | 0.74 (0.47–1.18) | 0.80 (0.42–1.52) | 0.71 (0.44–1.14) | 0.72 (0.38–1.38) |
| Age (years) (*n* = 622) | | | | |
| 16–34 | 1.00 | 1.00 | 1.00 | 1.00 |
| 35–44 | 1.53 (0.85–2.75) | **2.71 (1.15–6.37)** | 1.54 (0.85–2.76) | **2.72 (1.16–6.39)** |
| 45–54 | **2.74 (1.54–4.90)** | **6.03 (2.62–13.89)** | **2.76 (1.54–4.94)** | **6.06 (2.62–14.00)** |
| 55–64 | **3.70 (1.60–8.54)** | **9.46 (3.33–26.84)** | **3.90 (1.68–9.07)** | **9.93 (3.47–28.42)** |
| Marital status (*n* = 622) | | | | |
| Married/Cohabiting | 1.00 | 1.00 | 1.00 | 1.00 |
| Single | 0.74 0.45–1.21 | 0.92 (0.47–1.79) | 1.16 (0.66–2.03) | **2.24 (1.02–4.92)** |
| Divorced/Separated/Widowed | 1.48 (0.70–3.12) | **2.79 (1.13–6.88)** | 1.13 (0.51–2.50) | 1.83 (0.73–4.57) |
| Ethnicity (*n* = 622) | | | | |
| White British | 1.00 | 1.00 | 1.00 | 1.00 |
| Black Caribbean & Black African | 0.83 (0.46–1.50) | 0.42 (0.15–1.16) | 0.94 (0.51–1.73) | 0.55 (0.19–1.55) |
| Other | 1.31 (0.78–2.20) | 1.27 (0.63–2.59) | 1.41 (0.83–2.40) | 1.47 (0.70–3.05) |
| Migrant status (*n* = 621) | | | | |
| Born in UK | 1.00 | 1.00 | 1.00 | 1.00 |
| 0–10 years in the UK | 0.69 (0.37–1.27) | 0.36 (0.12–1.06) | 0.77 (0.41–1.44) | 0.47 (0.16–1.38) |
| 11 or more years in the UK | 1.45 (0.82–2.55) | 1.50 (0.68–3.31) | 1.18 (0.64–2.16) | 1.05 (0.44–2.49) |
| **Socioeconomic variables** | | | | |
| Housing Tenure (*n* = 621) | | | | |
| Own/ mortgage | 1.00 | 1.00 | 1.00 | 1.00 |
| Rent/part rent | 1.41 (0.86–2.31) | 0.73 (0.38–1.40) | **1.73 (1.03–2.90)** | 1.02 (0.50–2.08) |
| Rent free | 0.67 (0.24–1.86) | 0.18 (0.24–1.38) | 1.08 (0.36–3.19 | 0.39(0.05–3.39) |
| Education (highest level attained) (n = 620) | | | | |
| Degree level or above | 1.00 | 1.00 | 1.00 | 1.00 |
| Up to A-Level | 1.37 (0.80–2.35) | 1.30 (0.61–2.77) | 1.64 (0.94–2.87) | 1.74 (0.82–3.71) |
| Up to GCSE | 1.27 (0.66–2.44) | 1.69 (0.75–3.81) | 1.53 (0.79–2.99) | 2.17 (0.95–4.94) |
| No qualifications | 2.52 (1.14–5.57) | 2.60 (0.88–7.68) | 2.15 (0.95–4.87) | 1.87 (0.61–5.70) |
| Household Income (gross, annual) (*n* = 621) | | | | |
| £31,495 or more | 1.00 | 1.00 | 1.00 | 1.00 |
| £12,098–31,494 | 1.58 (0.92–2.71) | 0.91 (0.34–2.47) | 1.60 (0.93–2.75) | 0.91(0.33–2.53) |
| £0–12,097 | 1.01 (0.50–2.04) | **2.35 (1.05–5.24)** | 1.07 (0.52–2.18) | **2.53 (1.11–5.80)** |
| Do not know | 1.00 (0.47–2.14) | 1.72 (0.69–4.26) | 1.30 (0.60–2.85) | **2.69 1.07–6.73)** |
| Employment (n = 618) | | | | |
| Employed Full time | 1.00 | 1.00 | 1.00 | 1.00 |
| Employed part time | 1.14 (0.63–2.1) | **3.06 (1.27–7.35)** | 1.09 (0.59–2.01) | **2.82 (1.12–7.10)** |
| Student | **0.44 (0.20–0.99)** | 1.36 (0.51–3.63) | 0.61 (0.26–1.42) | 2.84 (0.93–8.73) |
| Unemployed | 1.64 (0.80–3.36) | **4.37 (1.56–12.22)** | 1.73 (0.84–3.58) | **4.83 (1.69–13.84)** |
| Early work exit | **4.08 (1.04–16.11)** | 33.84 (9.14–125.29) | 2.99 (0.69–12.94) | **16.86 (3.99–71.30)** |
| At home | 1.87 (0.73–4.79) | 3.34 (0.84–13.23) | 1.63 (0.36–4.26) | 2.91 (0.70–12.13) |
| Benefit receipt (n = 618) | | | | |
| No benefits | 1.00 | 1.00 | 1.00 | 1.00 |
| Benefits | **1.71 (1.01–2.87)** | **2.19 (1.05–4.55)** | 1.58 (0.93–2.69) | 2.03 (0.93–4.42) |

*(Continued)*

**Table 2.** (Continued)

| | Unadjusted | | Adjusted | |
|---|---|---|---|---|
| | 0 to ≥1 LTCs[b] | 1-to-Many LTCs[c] | 0 to ≥1 LTCs[b] | 1-to-Many LTCs[c] |
| | RRR (95% CI) | RRR (95% CI) | RRR (95% CI) | RRR (95% CI) |
| **Psychosocial variables** | | | | |
| Size of social network (n = **622**) | | | | |
| ≥5 | 1.00 | 1.00 | 1.00 | 1.00 |
| 3–4 | 1.38 (0.82–2.32) | 1.32 (0.62–2.79) | 1.50 (0.87–2.56) | 1.52 (0.71–3.26) |
| ≤2 | **3.46 (1.46–8.24)** | **5.42 (1.94–15.13)** | **3.31 (1.38–7.95)** | **5.02 1.69–14.94)** |
| Stressful life events (n = **622**) | | | | |
| ≤2 | 1.00 | 1.00 | 1.00 | 1.00 |
| 3–5 | 1.20 (0.72–2.00) | 2.17 (0.94–5.01) | 1.19 (0.70–2.02) | 2.05 (0.88–4.81) |
| ≥6 | **2.10 (1.16–3.81)** | **4.54 (1.86–11.07)** | **1.88 (1.01–3.51)** | **3.49 (1.44–8.46)** |
| **Health risk factors** | | | | |
| Body Mass Index (n = **609**) | | | | |
| Normal weight | 1.00 | 1.00 | 1.00 | 1.00 |
| Overweight | **2.06 (1.24–3.42)** | 1.01 (0.51–1.99) | **1.88 (1.12–3.16)** | 0.79 (0.39–1.58) |
| Obese | **2.03 (1.09–3.78)** | 0.95 (0.39–2.35) | 1.76 (0.94–3.28) | 0.71 (0.29–1.70) |
| Smoking status (n = **622**) | | | | |
| Non-smoker | 1.00 | 1.00 | 1.00 | 1.00 |
| Current Smoker | **1.80 (1.1–2.93)** | 1.64 (0.82–3.27) | **1.91 (1.16–3.14)** | 1.74 (0.83–3.66) |
| Alcohol Use (n = **622**) | | | | |
| Non-drinkers | 1.00 | 1.00 | 1.00 | 1.00 |
| Moderate drinkers | 1.22 (0.65–2.30) | 1.08 (0.4–2.90) | 1.12 (0.59–2.11) | 0.90 (0.33–2.47) |
| Hazardous alcohol use | 1.31 (0.62–2.78) | 1.71 (0.58–5.07) | 1.43 (0.65–3.11) | 1.93 (0.59–6.31) |
| **Health-related quality of life** | | | | |
| Self-rated health (*n* = **622**) | | | | |
| Excellent/very good | 1.00 | 1.00 | 1.00 | 1.00 |
| Good | **2.10 (1.28–3.44)** | **4.08 (1.84–9.09)** | **2.22 (1.33–3.68)** | **4.44 (1.99–9.91)** |
| Fair/Poor | **4.03 (1.93–8.44)** | **16.59 (6.89–39.95)** | **4.34 (2.04–9.23)** | **18.70 (7.71–45.35)** |
| Functional impairment due to physical health (n = **622**) | | | | |
| No | 1.00 | 1.00 | 1.00 | 1.00 |
| Yes | **2.22 (1.09–4.52)** | **5.66 (2.63–12.17)** | **2.20 (1.05–4.62)** | **5.61 (2.47–12.75)** |
| Functional impairment due to emotional health (*n* = **622**) | | | | |
| No | 1.00 | 1.00 | 1.00 | 1.00 |
| Yes | **2.51 (1.39–4.52)** | **2.56 (1.11–5.86)** | **2.76 (1.50–5.08)** | **3.10 (1.25–7.69)** |
| **Somatic symptoms** | | | | |
| Somatic symptom severity (*n* = **621**) | | | | |
| Low (0/4) | 1.00 | 1.00 | 1.00 | 1.00 |
| Medium (5/9) | **1.70 (1.01–2.87)** | **3.66 (1.90–7.03)** | 1.69 (1.00–2.87) | **3.76 (1.85–7.63)** |
| High (≥10) | **3.63 (1.74–7.60)** | **4.98 (1.87–13.28)** | **4.35 (2.03–9.32)** | **7.49 (2.61–21.46)** |

S1 and S2 represent the two SELCoH sample surveys at timepoints 1 and 2, respectively. Relative risk, RRR, Is weighted to account for survey design. "No reported LTC at either S1 or S2" represents the reference category in the multinomial regressions. All variables are measured at the S1 timepoint except ethnicity which was only measured at the S2 timepoint. CI = 95% confidence interval for the relative risk ratio. Bold indicates statistical significance. There were no participants in the underweight group in either the initiation or progression trajectory categories, so the underweight category is not shown in the table.

[a]Adjusted by age (continuous) and gender
[b] 0-to-≥ 1 LTC: S1 = 0 LTC, S2 ≥1 LTCs
[c] 1-to-MLTCs: S1 = 1 LTC, S2 ≥2 LTCs

number of stressful life events ($\geq 6$) at baseline, increased the risk of developing initial LTCs and also progression from one-to-many LTCs (Table 2). Health risk factors included being overweight and smoking which were both determinants of an increased risk of developing initial LTCs. Whilst there was an increased risk of developing initial LTCs in those who were obese (RRR 1.76, 95% CI 0.94–3.28), and also of progressing from one-to-many LTCs in those who were smokers (RRR 1.74, 95% CI 0.83–3.66), neither of these were statistically significant.

Baseline measures related to health-related quality of life (self-rated health and functional impairment due to both physical and emotional health) and somatic symptom severity, were all determinants of both LTC initiation and progression, with poor/fair self-rated health at baseline having the strongest association with progressing from one-to-many LTCs (RRR 18.70, 95% CI 7.71–45.35).

Socioeconomic determinants of negative health transitions included housing tenure, income and employment status (Table 2). After adjusting for age and gender, renting/part-renting a property was associated with an increased the risk of developing initial LTCs, and having a low income or not knowing the household income, were both associated with an increased risk of progression from one-to-many LTCs. Employment status at baseline increased the risk of progressing from one-to-many LTCs for those who were employed part-time, unemployed or having taken an early work exit, compared to those who were employed full-time.

## LTCs and employment

The average number of LTCs at follow-up was greatest in the group that had made an early work exit (2.2 LTCs), with the full-time employed and student group reporting the lowest average number of LTCs (0.5 and 0.4 LTCs respectively). Compared to the full-time employee group, those who were employed part-time and the unemployed both had a higher average number of LTCs (0.6 and 0.8 respectively). Those who were at home also had an average of 0.6 LTCs. The distribution of the eight most prevalent LTCs within each employment category at follow-up (S2) is shown in Table 3.

**Table 3. Frequency and prevalence distribution of LTCs within each employment category at follow-up.**

| | | Employed FT (n = 431) | Employed PT (n = 161) | Unemployed (n = 90) | Early work exit (n = 78) | Student (n = 82) | At home (n = 51) |
|---|---|---|---|---|---|---|---|
| | N | n (%) | n (%) | n (%) | n (%) | n (%) | n (%) |
| Asthma | 82 | 34 (7.5) | 13 (9.3) | 9 (9.2) | 19 (24.3) | 5 (5.6) | 2 (4.2) |
| Depression | 79 | 16 (3.5) | 11 (6.4) | 12 (12.7) | 29 (37.8) | 9 (9.7) | 2 (3.7) |
| Diabetes | 30 | 8 (1.5) | 5 (2.9) | 5 (4.6) | 9 (11.1) | 1 (0.8) | 2 (2.9) |
| Gastrointestinal conditions | 38 | 13 (2.8) | 4 (2.2) | 7 (7.9) | 7 (9.9) | 7 (8.2) | 0 |
| Gynaecological conditions | 26 | 9 (2.1) | 5 (2.9) | 2 (1.7) | 4 (5.0) | 4 (4.1) | 2 (2.8) |
| High blood pressure | 81 | 31 (6.2) | 12 (6.2) | 8 (7.2) | 26 (33.1) | 0 | 4 (6.3) |
| Migraine | 34 | 14 (3.1) | 5 (3.3) | 6 (6.9) | 7 (10.3) | 1 (1.1) | 1 (1.8) |
| Musculoskeletal conditions | 106 | 37 (7.5) | 20 (10.4) | 8 (7.8)) | 36 (47.1) | 1 (1.6) | 4 (6.7) |

S2 represents data collected at the follow-up SELCoH interviews. Frequencies are unweighted. Percentages are weighted to account for survey design. Column percentages may add up to >100% for each employment category due to multimorbidity. N = 893 and includes all follow-up data.

Musculoskeletal conditions were the most prevalent condition in those who were employed full-time (7.5%), part-time (10.4%), the early work exit (41.7%) and at home (6.7%) groups, with asthma additionally being the most prevalent in the full-time employee group (7.5%). In contrast, depression was the most prevalent condition in those who were unemployed (12.7%) and students (9.7%). The early work exit group (which includes permanently sick, disabled and the early-retired), had the highest overall prevalence of LTCs. The four most prevalent conditions in this group were musculoskeletal conditions, depression, high blood pressure and asthma, which were also the most prevalent conditions in the employed full-time and part-time groups. Of those who were economically active (employed or unemployed but available for work), depression was reported by 12.7% of the unemployed group compared to 4% and 6% of the full-time and part-time employee groups respectively.

## Discussion

This study provides evidence that the journey to multimorbidity may have both common and unique risk factors according to the stage of MLTC progression. At the early stage of developing initial LTCs, health behaviours appear to be important, with smoking, being overweight and renting being unique to this initial stage. In contrast, the progression from one to multiple LTCs was associated with socioeconomic factors including low household income and the three employment categories of being employed part-time, being unemployed or having made an early work exit. This could be explained by the fundamental social cause theory which states that those with access to more and better quality socioeconomic resources (such as, better employment circumstances and income) and beneficial social connections, will have a lower mortality risk due to increased capacity to manage their health [12]. Employers may also provide specific health resources such as health screening in workplaces and health insurance [8, 12]. Having an income as a result of being employed also facilitates participation in society (and thereby supporting social support networks), with the additional psychological benefits of providing a secure social context and a sense of identity [8]. In contrast, both unemployment and part-time employment, were linked to negative health transitions (specifically, progressing from one to multiple LTCs). Whilst unemployment is known to be associated with poorer physical and mental health and is a risk factor for negative health transitions [8, 22], the role of part-time employment in negative health transitions is less clear. One possibility is that there is a higher proportion of people with one LTC at baseline who found the need to change from full-time to part-time employment, and so this group may already be at increased risk. The link between unemployment and poor mental health was supported by our findings that whilst depression was the most prevalent LTC in the unemployed group, it was the fourth most prevalent condition in those who were employed full-time and the third most prevalent condition in those who were employed part-time. Although musculoskeletal conditions and asthma were the two most prevalent physical conditions in those who were employed both full- and part-time, they were also the most prevalent across the whole sample irrespective of employment status, which may reflect a generally high occurrence in working age adults in this inner-city cohort.

Whilst the risk of progression from good health to multimorbidity increased with age, this study showed that even in the younger age group (35 to 44 years), there was a greater (and statistically significant) risk of progressing from one to multiple LTCs, even when compared to developing a first LTC. This may be linked to socioeconomic factors that were also found to increase the risk of progression to multimorbidity. Previous evidence for multimorbidity in younger age groups has been shown to occur in areas of high socioeconomic deprivation where the prevalence of multimorbidity in young and middle-aged adults living in

socioeconomically deprived areas was equivalent to those 10–15 years older living in more affluent areas [6]. Since the present study uses data from an inner-city population known to have high levels of socioeconomic deprivation overall [23, 24], the specific socioeconomic factors of low household income and not being employed full-time being linked to an earlier progression to multimorbidity, provides more detailed supporting evidence.

It is of interest that whilst there may be different risk factors linked to LTC initiation and then progression to multimorbidity, there were common factors such as small social network size, increased number of stressful life events, poor/fair self-rated health, increased functional impairments due to physical and mental health, and increased somatic symptom severity. Social networks are linked to social support, with the latter being defined as the subjective appraisal of being able to draw on the resources of a social network [25]. Insufficient social support has been linked to higher rates of physical and mental health conditions, and in the present study, if smaller social networks decreased the opportunity for social support, this may have increased the risk of developing LTCs [26]. Similarly, higher levels of perceived social support have been shown to have a positive impact on health-related quality of life in patients with both single and multiple LTCs [25, 27].

A strength of this study is its longitudinal design with the independent variables being measured at baseline (S1) indicating potential risk factors for the two negative health transition trajectories. Additionally, variables relating to health perceptions (self-rated health, functional impairment due to both physical and mental health and somatic symptom severity) were measured prior to the increased prevalence of LTCs measured at follow-up (S2) for each of the trajectories. If negative health perceptions at S1 are associated with the development of initial LTCs at S2, the onset of negative health perceptions may represent an early stage of the diagnostic journey between the onset of symptoms and a formal diagnosis of an initial LTC. Since this study showed an increased risk of developing an initial LTC in those who were overweight and smokers, a better understanding of the socioeconomic and behavioural profile of individuals with early self-reported negative health perceptions may assist in designing early targeted interventions.

This study is limited by the small sample size and would benefit from repeating in a larger population to confirm the results, and carry out comparisons in a wider setting including rural communities. In contrast to many studies using patient records, all self-reported conditions were included in this study. However, a larger sample size would have enabled a larger number of LTCs to be analysed separately rather than being combined into similar condition groups. Despite the advantages of including all self-reported conditions, it should also be noted that without access to medical records, we were unable to confirm whether a formal diagnosis had been received for all conditions, or if there were un-reported conditions that had not yet received a diagnosis or were being successfully managed by the individual. A larger sample size and the addition of medical record data would also enable factors supporting positive health transitions to be investigated. By expanding the scope of future studies to include positive health transitions, this would provide guidance for health promotion interventions aimed at slowing the journey to multimorbidity. It should also be acknowledged that whilst this study is longitudinal in design with two timepoints, the testing of baseline factors that increase the likelihood of the two negative LTC trajectories should be interpreted with caution, and can only imply causality rather than prove it. Also, reverse causality cannot be excluded, particularly in the case of employment where health status may lead to reduced working hours or unemployment.

In conclusion, this study demonstrates that the journey to multimorbidity is complex, and suggests there are both common and unique factors linked to the two stages of initiation and progression of LTCs. Further research focusing on developing health promotion interventions

in younger, healthy participants, should aim to increase understanding of the role of precursors to LTC in the most prevalent functionally impairing LTCs that impact the ability to engage in employment. Whilst we have shown that not being employed full-time may have a negative influence on health, future research should consider the quality of employment (employment conditions and relationships) and job characteristics (the nature of the job). Investigation of a wide range of employment characteristics could elaborate on specific causal relationships between employment and the development of multiple long-term health conditions.

## Author Contributions

**Conceptualization:** Ira Madan, Sharon A. M. Stevelink.

**Formal analysis:** Anne L. Stagg.

**Funding acquisition:** Ira Madan, Sharon A. M. Stevelink.

**Investigation:** Anne L. Stagg.

**Methodology:** Lisa Harber-Aschan, Stephani L. Hatch, Nicola T. Fear, Ira Madan, Sharon A. M. Stevelink.

**Project administration:** Anne L. Stagg.

**Visualization:** Anne L. Stagg.

**Writing – original draft:** Anne L. Stagg.

**Writing – review & editing:** Anne L. Stagg, Lisa Harber-Aschan, Stephani L. Hatch, Nicola T. Fear, Sarah Dorrington, Ira Madan, Sharon A. M. Stevelink.

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
