## [Decision Letter · Decision Letter 0]

4 Apr 2023

PONE-D-22-33742Risk factors for the progression to multimorbidity among UK urban working-age adults. A community cohort studyPLOS ONE

Dear Dr. Stagg,

Thank you for submitting your manuscript to PLOS ONE. After careful consideration, we feel that it has merit but does not fully meet PLOS ONE’s publication criteria as it currently stands. Therefore, we invite you to submit a revised version of the manuscript that addresses the points raised during the review process.

 In particular, the analysis requires extensive and careful revision as pointed out by reviewer 1. Please ensure that your decision is justified on PLOS ONE’s publication criteria and not, for example, on novelty or perceived impact.

We look forward to receiving your revised manuscript.

Kind regards,

Andrea Dell'Isola

Academic Editor

PLOS ONE

Journal Requirements:

"This paper represents independent research part funded by NIHR Maudsley Biomedical Research Centre at South London and Maudsley NHS Foundation Trust and King’s College London, and the NIHR. The views expressed are those of the authors and not necessarily those of the NHS, the NIHR or the Department of Health and Social Care."

3. We noted in your submission details that a portion of your manuscript may have been presented or published elsewhere. "The supporting information table listing the prevalence of health conditions at both timepoints has been published in a preliminary study which has recently been published in BMJ Open.

However, it is also key information for this paper in relation to the number of individual long-term conditions at baseline and at follow-up as it was used to create the long-term condition variables showing acquisition of a first LTC between the two timepoints, and also development of multimorbidity over the same timeframe. Since it has been published before, it was included only as a supplementary table so readers could easily refer to it." Please clarify whether this [conference proceeding or publication] was peer-reviewed and formally published. If this work was previously peer-reviewed and published, in the cover letter please provide the reason that this work does not constitute dual publication and should be included in the current manuscript.

Reviewers' comments:

Reviewer's Responses to Questions

**Comments to the Author**

1. Is the manuscript technically sound, and do the data support the conclusions?

Reviewer #1: Yes

Reviewer #2: Yes

2. Has the statistical analysis been performed appropriately and rigorously? 

Reviewer #1: No

Reviewer #2: Yes

3. Have the authors made all data underlying the findings in their manuscript fully available?

Reviewer #1: Yes

Reviewer #2: Yes

4. Is the manuscript presented in an intelligible fashion and written in standard English?

Reviewer #1: Yes

Reviewer #2: Yes

5. Review Comments to the Author

Reviewer #1: This is an interesting and well-written paper. Its strengths include the longitudinal data and the data on employment - which is particularly valuable in the context of multimorbidity among working-age people. Specific comments below.

1. The model description is not 100% clear. It would be good to know the parameterisation and any assumptions. I also wonder if it would be easier to run two separate logistic regression models. One for any LTC and then another, conditional on having a LTC what is the odds of developing further LTCs. This would make it difficult to model improvement, but the trade off is it would be simpler.

2. There seem to be a lot of parameters for the relatively small number of events (I counted 39 even though there were only 46 people who transitioned from 1 to many LTCs). I know that the rule of thumb 10 events per variable is an overstatement, but 39 still seems excessive. Moreover, age could be modelled using splines or fractional polynomials to reduce the number of parameters that need to be fitted.

3. Related to this, if I understand correctly these coefficients are all mutually adjusted (even for the unadjusted model). This leads to the table 2 fallacy (https://pubmed.ncbi.nlm.nih.gov/23371353/). Instead it would be better to draw some DAGS and decide on which covariates should go into models asking questions which are relevant for clinical practice or for determining policy.

4. Notwithstanding the model fitting some post-processing to obtain absolute risks of transitioning to each state for people with different covariate values would be useful. The odds ratio scale is fine for relative effects, but less good for describing patterns of disease and risks, which is the purpose here.

5. It would be useful to discuss why people were not followed-up in the second round. Could this be due to ill health preventing them from completing the wave? It would be good to report the number and characteristics of people who were invited for a second wave but did not participate.

6. In the discussion the statement is made that the odds ratios are different for 0/1 versus 1/many LTCs. If this is to be asserted the difference in the odds ratios should be estimated. Again, however, some thought needs to be given to scale. Some characteristics may increase risk MORE or LESS on the absolute scale while being the same on the relative scale (and vice versa).

7. Would be good to see some numbers in the abstract results section.

8. The introduction states that having a single LTC occurs before multiple LTCs - isn't this almost true by definition (at least for sufficiently granular and complete data)?

9. nice to see the number of events in table 1. Would be good to see this at the top of Table 2 also.

10. Not sure of the usefulness of the unadjusted coefficients in table 2

11. Would be good to share code and (if possible) aggregate level data (I know IPD cannot be shared).

Reviewer #2: The manuscript “Risk factors for the progression to multimorbidity among UK urban working-age adults. A community cohort study” explores factors associated with the initiation and progression of long-term chronic conditions among the working population in the UK. This is a very nice paper with an important contribution to the literature. The manuscript is well written and organized. The methodology and analytical methods used are appropriate for the study goals and results are presented with clarity. I only have minor questions/clarifications as follows.

• Page 6, line 116-124: The analytical sample included participants who completed both surveys. This is understandable given the objectives. However, I wonder if not responding at wave 2 may partially related to chronic conditions or some important participant characteristics. Could the authors comment on the differences in # of conditions and characteristics of those excluded due to loss to follow up? I think a brief comment in the discussion/limitations should suffice.

• Page 10, lines 204-210: It is not clear how adjusted models were fit. For example, for BMI, the adjusted model included BMI, age, sex only or the RRR is from a model with all variables? Please clarify. If the latter, please comment on the possible impact of collinearity to the results.

6. PLOS authors have the option to publish the peer review history of their article (what does this mean?). If published, this will include your full peer review and any attached files.

Reviewer #1: **Yes: **David A McAllister

Reviewer #2: No

---

## [Author Response · Author response to Decision Letter 0]

1 Jun 2023

1 Editor comments / Journal requirements 

Please ensure that your manuscript meets PLOS ONE's style requirements, including those for file naming. The PLOS ONE style templates can be found at https://journals.plos.org/plosone/s/file?id=wjVg/PLOSOne_formatting_sample_main_body.pdf and

Response from authors:

We have checked PLOS ONE’s style requirements are applied throughout the document and one adjustment has been made: 

 1) Table 2 legend has been altered so it is in one section with the footnotes following the legend.

 2) There is no longer a supplementary table, so the re-naming of that table is no longer required.

 3) Please could you ensure that Prof Ira Madan and Dr Sharon Stevelink are joint last/senior authors. Thank you.

2 Editor comments / Journal requirements 

Thank you for stating the following in the Competing Interests section:

"This paper represents independent research part funded by NIHR Maudsley Biomedical Research Centre at South London and Maudsley NHS Foundation Trust and King’s College London, and the NIHR. The views expressed are those of the authors and not necessarily those of the NHS, the NIHR or the Department of Health and Social Care."

Response from authors:

Please could you update our Competing Interests statement with the following statement which includes the reference to PLOS ONE policies on sharing data and materials: 

“This paper represents independent research part funded by NIHR Maudsley Biomedical Research Centre at South London and Maudsley NHS Foundation Trust and King’s College London, and the NIHR. The views expressed are those of the authors and not necessarily those of the NHS, the NIHR or the Department of Health and Social Care. This does not alter our adherence to PLOS ONE policies on sharing data and materials.”

3 Editor comments / Journal requirements 

We noted in your submission details that a portion of your manuscript may have been presented or published elsewhere. "The supporting information table listing the prevalence of health conditions at both timepoints has been published in a preliminary study which has recently been published in BMJ Open.

However, it is also key information for this paper in relation to the number of individual long-term conditions at baseline and at follow-up as it was used to create the long-term condition variables showing acquisition of a first LTC between the two timepoints, and also development of multimorbidity over the same timeframe. Since it has been published before, it was included only as a supplementary table so readers could easily refer to it." Please clarify whether this [conference proceeding or publication] was peer-reviewed and formally published. If this work was previously peer-reviewed and published, in the cover letter please provide the reason that this work does not constitute dual publication and should be included in the current manuscript. 

Response from authors:

We can confirm that the supplementary Table S1 has been published in the following peer-reviewed preliminary study: Stagg AL, Hatch S, Fear NT, et al. Long-term Health conditions in UK working-age adults: a cross-sectional analysis of associations with demographic, socioeconomic, psychosocial and health-related factors in an inner-city population. BMJ Open 2022;12:e062115. doi:10.1136/bmjopen-2022-062115 

The reason this table was included as a supplementary table was as previously stated, for ease of access. However, this has now been amended so that reference is now made to the initial publication where readers can access the full information about the numbers of conditions at both timepoints (see Revised Manuscript with Track Changes, page 7, lines 140 and 141). The original supplementary file is therefore no longer necessary. The supporting information section on page 27, lines 506-517 has also been removed.

4 Editor comments / Journal requirements 

In your Data Availability statement, you have not specified where the minimal data set underlying the results described in your manuscript can be found. PLOS defines a study's minimal data set as the underlying data used to reach the conclusions drawn in the manuscript and any additional data required to replicate the reported study findings in their entirety. All PLOS journals require that the minimal data set be made fully available. For more information about our data policy, please see http://journals.plos.org/plosone/s/data-availability.

Response from authors on Data Availability:

This study used third party data from the SELCoH survey. The following data availability statement should be included: 

“SELCoH data is available to researchers upon request. Please contact selcoh@kcl.ac.uk to apply for access. For more information visit www.kcl.ac.uk/research/selcoh”

Further Editor comments / Journal requirements (Email: 26 05 23)

1 and 2 Editor comments / Journal requirements 

We note that your uploaded Cover Letter is not a Cover Letter but a Response to Reviewers file. It is important that you include a cover letter with your manuscript. Please ensure that this letter is addressed specifically to PLoS ONE. Please also include

* why this manuscript is suitable for publication in PLoS ONE.

* how does your paper provide a worthwhile addition to the scientific literature?

* how does your paper relate to previously published work?

* which types of scientists do you believe will be most interested in your study?

Please also note that the Cover Letter and Response to Reviewers file should be uploaded as two separate files.

2. Please upload a Response to Reviewers letter which should include a point by point response to each of the points made by the Editor and / or Reviewers. (This should be uploaded as a 'Response to Reviewers' file type.) Please follow this link for more information: https://eur03.safelinks.protection.outlook.com/?url=http%3A%2F%2Fblogs.plos.org%2Feveryone%2F2011%2F05%2F10%2Fhow-to-submit-your-revised-manuscript%2F&data=05%7C01%7Canne.l.stagg%40kcl.ac.uk%7C0cb97bc2fa5d4fcb1fb908db5d96cff3%7C8370cf1416f34c16b83c724071654356%7C0%7C0%7C638206674534014930%7CUnknown%7CTWFpbGZsb3d8eyJWIjoiMC4wLjAwMDAiLCJQIjoiV2luMzIiLCJBTiI6Ik1haWwiLCJXVCI6Mn0%3D%7C3000%7C%7C%7C&sdata=upX54CLXyfauygwUSl3vmgiwyj368IvaEQrw5KAAuCY%3D&reserved=0

Response from authors:

The cover letter and Response to Reviewers letters have now been uploaded as two separate files. 

3 Editor comments / Journal requirements 

Your ethics statement should only appear in the Methods section of your manuscript. If your ethics statement is written in any section besides the Methods, please move it to the Methods section and delete it from any other section. Please ensure that your ethics statement is included in your manuscript, as the ethics statement entered into the online submission form will not be published alongside your manuscript.

Response from authors:

The ethics statement has now been moved within the Methods from the Statistical analyses section to the Study design, setting and participants section (see Revised Manuscript with Track Changes page 6 lines 125-128). Thank you for pointing this out as this is a more appropriate section for the ethics statement which is linked to the SELCoH data collection.

4 Editor comments / Journal requirements 

We note your current Data Availability Statement: "This study used third party data from the SELCoH survey. The following data availability statement should be included:

SELCoH data is available to researchers upon request. Please contact selcoh@kcl.ac.uk to apply for access. For more information visit https://www.kcl.ac.uk/research/selcoh

Regarding third party data, please confirm whether or not the authors received any special privileges in accessing the data that other researchers would not have.

Response from authors:

We can confirm that the authors did not receive any special privileges in accessing the data that other researchers would not have.

1 Reviewer 1 Comments to the Author 

The model description is not 100% clear. It would be good to know the parameterisation and any assumptions. I also wonder if it would be easier to run two separate logistic regression models. One for any LTC and then another, conditional on having a LTC what is the odds of developing further LTCs. This would make it difficult to model improvement, but the trade off is it would be simpler 

Response from authors:

We agree with the reviewer model description should be made clearer, and a similar comment on clarity was also raised by the second reviewer. To address this, the model description in the transcript has been revised in the statistical analyses section to make it clearer (see document ‘Revised Manuscript with Track Changes’, page 10 lines 209-217) and page 14 lines 253-254. In explanation to the reviewers, we used multinomial logistic regressions and with the dependent outcome variable consisting of the following categories: (1) the odds of developing any number of LTCs at the second timepoint having not had any LTCs at baseline, and (2) when there is one LTC at baseline, the odds of developing further LTCs by the second timepoint. (3) No reported LTCs at either S1 or S2 was used as the reference category in the regressions. The outcome is therefore similar to the two models the reviewer suggests but has the advantage of including the 479 individuals in the healthy group who had no LTCs at either timepoint (the reference group), is that it increases the strength of the analysis compared to running two separate regression models where we would lose data from these 479 individuals. Since we note that the second reviewer was satisfied with the statistics used, we have not changed our modelling, but are happy to reconsider if the editor believes that re-modelling the analysis is essential.

2 Reviewer 1 Comments to the Author 

There seem to be a lot of parameters for the relatively small number of events (I counted 39 even though there were only 46 people who transitioned from 1 to many LTCs). I know that the rule of thumb 10 events per variable is an overstatement, but 39 still seems excessive. Moreover, age could be modelled using splines or fractional polynomials to reduce the number of parameters that need to be fitted. 

Response from authors:

This comment regarding the balance between the number of parameters for the number of participants appears to be linked to the previous comment about lack of clarity over the model description as it would be highly relevant where all the predictor variables were tested simultaneously within in the same model. However, we did not carry out a multiple regression, and each predictor variable was tested independently for the multinomial LTC outcomes, apart from the adjusted model which included age and gender, so a maximum of 3 predictor variables in the adjusted model.

3 Reviewer 1 Comments to the Author 

Related to this, if I understand correctly these coefficients are all mutually adjusted (even for the unadjusted model). This leads to the table 2 fallacy (https://pubmed.ncbi.nlm.nih.gov/23371353/). Instead it would be better to draw some DAGS and decide on which covariates should go into models asking questions which are relevant for clinical practice or for determining policy. 

Response from authors:

This comment has been addressed in the response to this reviewer’s the first comment as it relates to the model description, which we have now clarified and explained that the coefficients are not all mutually adjusted.

4 Reviewer 1 Comments to the Author 

Notwithstanding the model fitting some post-processing to obtain absolute risks of transitioning to each state for people with different covariate values would be useful. The odds ratio scale is fine for relative effects, but less good for describing patterns of disease and risks, which is the purpose here.

Response from authors:

Again, we think that this comment is linked to the lack of clarity about the regression modelling we used. We hope that this has now been addressed with the revisions that we have made.

5 Reviewer 1 Comments to the Author 

It would be useful to discuss why people were not followed-up in the second round. Could this be due to ill health preventing them from completing the wave? It would be good to report the number and characteristics of people who were invited for a second wave but did not participate. 

Response from authors:

The second reviewer has also raised a similar comment about reasons for lack of response at follow-up particularly in relation to ill health, which we have addressed below in response to their comment.

6 Reviewer 1 Comments to the Author 

In the discussion the statement is made that the odds ratios are different for 0/1 versus 1/many LTCs. If this is to be asserted the difference in the odds ratios should be estimated. Again, however, some thought needs to be given to scale. Some characteristics may increase risk MORE or LESS on the absolute scale while being the same on the relative scale (and vice versa). 

Response from authors:

We were a little uncertain as to the interpretation of this comment, but assume that it is in reference to the following sentence (page 19, lines 333-336): “Whilst the risk of progression from good health to multimorbidity increased with age, this study showed that even in the younger age group (35 to 44 years), there was a greater risk of progressing from one to multiple LTCs, even when compared to developing a first LTC.”

 This sentence primarily relates to the greater risk of progressing from one to multiple LTCs at the younger age (35-44 years) group which was shown to be statistically significant, whilst the risk of the initial development from none to one-or-more LTCs in the same age group was not statistically significant in both the unadjusted and adjusted (by gender) models. In addition, the relative risk ratios increase as age increases and this indicates an increased risk of developing LTCs with age. It may be that the reviewer was again considering the model to be a multiple logistic regression, which has now been clarified to not be the case.

 We have added in a comment on page 19 line 334-335 that this increased risk is statistically significant. However, if this is not the correct interpretation of the reviewer’s comment, please could you provide more clarification so that we can make the appropriate revision. 

7 Reviewer 1 Comments to the Author 

Would be good to see some numbers in the abstract results section. 

Response from authors:

The authors would have like to have a included some numbers in the abstract section, but the word count limitation prevented this.

8 Reviewer 1 Comments to the Author 

The introduction states that having a single LTC occurs before multiple LTCs - isn't this almost true by definition (at least for sufficiently granular and complete data)? 

Response from authors:

The authors agree with the comment that the development of multiple LTCs from 1 LTC is a common fact. The reason it is included in the introduction is because the paper is looking at risk factors for progressing from one to MLTCs, so this comment was made as the initial part of the argument justifying this study which identifies risk factors linked to this progression.

9 Reviewer 1 Comments to the Author 

nice to see the number of events in table 1. Would be good to see this at the top of Table 2 also. 

Response from authors:

If the numbers for Table 2 were put at the top of the table, it would not show the reference group of 479 participants, so we have inserted the numbers against each of the indicator variables (pages 14-16). This also then indicates where the small number of missing records lie (see also the comment about missing data on page 9, line 201-202 in the document ‘Revised Manuscript with Track Changes’).

10 Reviewer 1 Comments to the Author 

Not sure of the usefulness of the unadjusted coefficients in table 2 

Response from authors:

This comment is indeed understandable if all the coefficients had been tested together, as they would not be unadjusted results. However, having revised the script to provide a clearer explanation of the modelling used, we hope that now the inclusion of the unadjusted values in this table makes sense since all the predictor variables were tested individually. The adjusted value columns then show results where each predictor variable was adjusted by age and gender so that a direct comparison can be made between the unadjusted and adjusted relative risk ratios.

11 Reviewer 1 Comments to the Author 

Would be good to share code and (if possible) aggregate level data (I know IPD cannot be shared).

Response from authors:

As acknowledged by the reviewer, individual participant data cannot be directly shared. However, access to the SELCoH data is available and details have now been provided (see the updated data availability statement). 

Whilst the entire coding process covers data cleaning and another preliminary study, so not all relevant for this paper, useful examples of the code used for the unadjusted and adjusted regressions (using marital status as an example) are:

svy:mlogit ltc_change_3gps i.relationship_s1,rrr

svy:mlogit ltc_change_3gps i.relationship_s1 i.sex age_s1cont ,rrr

This code should also help to clarify for the reviewers the modelling we used for the regressions.

1 Reviewer 2 Comments to the Author 

Page 6, line 116-124: The analytical sample included participants who completed both surveys. This is understandable given the objectives. However, I wonder if not responding at wave 2 may partially related to chronic conditions or some important participant characteristics. Could the authors comment on the differences in # of conditions and characteristics of those excluded due to loss to follow up? I think a brief comment in the discussion/limitations should suffice. 

Response from authors:

This is an interesting comment, and whilst the cleaned data set used in this study did not include this group, the authors have investigated the SELCoH data for health-related information about those who did not respond at follow-up. 21 participants were ineligible for the second wave due to death or poor health. However, the data used for the current study was restricted to participants under the age of 64 at follow-up as it focused on working-age participants, and none of these 21 participants were in this age group. Consequently, we would be unable to comment further on this as an actual limitation within our study.

2 Reviewer 2 Comments to the Author 

Page 10, lines 204-210: It is not clear how adjusted models were fit. For example, for BMI, the adjusted model included BMI, age, sex only or the RRR is from a model with all variables? Please clarify. If the latter, please comment on the possible impact of collinearity to the results. 

Response from authors:

The comment by this reviewer highlighting the need for clarity regarding the modelling has been addressed in the response to the first reviewer who also raised this (please see the response to reviewer 1, comment 1). The description in the methods section has now been altered (see Revised Manuscript with Track Changes page 10 lines 209-217). In summary, the multinomial LTC trajectory variable was regressed on each indicator variable individually. Two models were tested 1) these regressions were carried out unadjusted and 2) then adjusted by gender and age.

---

## [Decision Letter · Decision Letter 1]

4 Jul 2023

PONE-D-22-33742R1Risk factors for the progression to multimorbidity among UK urban working-age adults. A community cohort studyPLOS ONE

Dear Dr. Stagg,

Thank you for submitting your manuscript to PLOS ONE. After careful consideration, we feel that it has merit but does not fully meet PLOS ONE’s publication criteria as it currently stands. Therefore, we invite you to submit a revised version of the manuscript that addresses the points raised during the review process.  Reviewer 2 was satisfied with your response while reviewer one was not available to review the manuscript. This prompted us to review your manuscript again. This further review highlighted some issues that were not pointed out during the previous round of review.

We look forward to receiving your revised manuscript.

Kind regards,

Andrea Dell'Isola

Academic Editor

PLOS ONE

Additional Editor Comments:

General: Please accommodate the request from Reviewer One to provide estimates in the abstract. Currently, the abstract contains excess information that could be condensed to make space for these required amendments.

We also ask that you revise your manuscript to refrain from using causal language (e.g., "determine"), as the present study design does not permit assertions of causality.

Introduction: We advise that you abbreviate the introduction to keep its focus consistent with the aim of the study. Given that the study's objective is exploratory, it appears excessive space is allocated to specific factors, such as occupation. Please shorten the introduction accordingly and realign it with the study's aims.

Methods: It is necessary to provide a clearer account of those individuals who did not participate in the second wave of the study and were subsequently excluded. We recommend including a flowchart as a supplementary file, outlining the participant trajectory. Moreover, we ask that you supply a table comparing the baseline characteristics of participants who met the inclusion criteria but did not participate in the follow-ups.

We noted that you group certain conditions together that are treated as a single condition in your analysis, as per Stagg et al. (2022). In this regard, please provide a rationale for the validity of classifying subjects as having a single condition when they may have multiple LTCs. Addressing this issue is crucial, as it may necessitate significant alterations in your analysis.

Your exclusion of individuals whose health status improves may introduce significant selection bias. The current emphasis on studying individuals with a deteriorating health status does not sufficiently justify your choice of analysis. We strongly suggest revising your analysis to include all participants.

Discussion: Please refine the structure of your discussion for improved readability. For instance, the correlation between employment categories is discussed at two different points (lines 318 and 338). Consolidating this discussion into a single paragraph would greatly enhance readability. Also, we recommend limiting speculative conclusions, as your study merely observes associations (e.g., employment can both be a potential causal factor for multimorbidity, or employment could reflect health status at baseline which is the real risk factor for multimorbidity).

Alo, please reconsider carefully your limitation statement. Consider for example the limited number of LTCs, the grouping of LTCs, the lack of information on propensity to seek care (and thus receive a diagnosis).

We appreciate your attention to these details and look forward to receiving your revised manuscript.

Reviewers' comments:

Reviewer's Responses to Questions

**Comments to the Author**

1. If the authors have adequately addressed your comments raised in a previous round of review and you feel that this manuscript is now acceptable for publication, you may indicate that here to bypass the “Comments to the Author” section, enter your conflict of interest statement in the “Confidential to Editor” section, and submit your "Accept" recommendation.

Reviewer #2: All comments have been addressed

2. Is the manuscript technically sound, and do the data support the conclusions?

Reviewer #2: Yes

3. Has the statistical analysis been performed appropriately and rigorously? 

Reviewer #2: Yes

4. Have the authors made all data underlying the findings in their manuscript fully available?

Reviewer #2: Yes

5. Is the manuscript presented in an intelligible fashion and written in standard English?

Reviewer #2: Yes

6. Review Comments to the Author

Reviewer #2: I have no further comments/suggestions. The authors have addressed all my questions and comments.

7. PLOS authors have the option to publish the peer review history of their article (what does this mean?). If published, this will include your full peer review and any attached files.

Reviewer #2: No

---

## [Author Response · Author response to Decision Letter 1]

17 Aug 2023

Response to the additional comments from the editors

Note: All references to the location of revisions made to the manuscript relate to the tracked changes version. 

1. Editor comment:

General: Please accommodate the request from Reviewer One to provide estimates in the abstract. Currently, the abstract contains excess information that could be condensed to make space for these required amendments.

Response from authors:

The abstract has been condensed and relative risk ratios (RRRs) with 95% confidence intervals have been provided for all the significant unique risk factors for both initiation and progression pathways (Abstract pages 3 and 4, lines 49-73). The abstract is still within the maximum 300 word count requirements. 

2. Editor comment:

We also ask that you revise your manuscript to refrain from using causal language (e.g., "determine"), as the present study design does not permit assertions of causality.

Response from authors:

The use of ‘determinants’ is related to the longitudinal design of the study which has two timepoints with the baseline variables being tested against two trajectories of risk of developing LTCs (This was acknowledged by Reviewer 1 in their comments to the authors who cited the longitudinal data as a strength of the study). However, it is acknowledged that there is an over-use of ‘determinants’ in the paper and a lack of caution in interpreting the results, so the following revisions have been made: 

Revisions so that there is the appropriate use of the term ‘variables’ rather than ‘determinants’: 

• Abstract, page 3 line 53: ‘determinants’ changed to ‘variables’ 

• Methods, page 7 line 149: The heading has been revised from ‘Determinant variables’ to ‘Independent variables’.

• Methods page 8 line 175: ‘determinants’ changed to ‘variables’ 

• Table 1 pages 11-14: ‘determinants’ changed to ‘variables’ and Table 2 (pages 14-17) has been changed to match the terminology used in Table 1. 

• Discussion: Page 22 line 397: ‘determinant variables’ has been amended to ‘independent variables’

• Discussion: Page 22, line 399 ‘determinants’ has been changed to ‘variables’. 

Additional changes:

• Abstract page 4 line 70: ‘determinants’ has been changed to 'risk factors’.

• Methods page 14 line 256: ‘Determinants’ has been changed to ‘Risk factors’

• Discussion page 19 lines 321-322: The following sentence has been changed from ‘This study indicates that the journey to multimorbidity has both common and unique determinants ..’ to ‘This study provides evidence that the journey to multimorbidity may have both common and unique risk factors..’ 

• Discussion page 22 lines 385-387: The following sentence has been changed from ‘Whilst there may be different determinants of LTC initiation and then progression to multimorbidity, it is of interest that there were common determinants of small social support networks …’ to ‘It is of interest that whilst there may be different risk factors linked to LTC initiation and then progression to multimorbidity, there were common factors such as small social support networks...’

• Discussion page 22 lines 402-403: The following sentence has been changed from ‘If negative health perceptions at S1 are determinants of developing initial LTCs at S2,..’ to ‘If negative health perceptions at S1 are associated with the development of initial LTCs at S2,..’

• Discussion page 23 lines 418-419: ‘determinants of’ has been changed to ‘factors supporting’.

• Discussion page 23 line 428: ‘determinants of’ has been changed to ‘factors linked to’.

Revisions highlighting caution in interpreting the results: 

• The insertion of the following two sentences in the discussion, page 23, lines 421-426: ’It should also be acknowledged that whilst this study is longitudinal in design with two timepoints, the testing of baseline determinants of the two negative LTC trajectories should be interpreted with caution, and can only imply causality rather than prove it. Also, reverse causality cannot be excluded, particularly in the case of employment where health status may lead to reduced working hours or unemployment.’ 

• Discussion page 23 lines 427-428: this sentence has been amended so the conclusions are not stated so strongly. It has been changed from: ‘In conclusion, this study demonstrates that the journey to multimorbidity is complex, with both common and unique determinants of the two stages of …’ to ‘In conclusion, this study demonstrates that the journey to multimorbidity is complex, and suggests there are both common and unique factors linked to the two stages of …’. 

• Discussion page 24, lines 432-33: ‘Whilst we have shown that not being employed full-time can have a negative influence on health, ..’ has been changed to ‘Whilst we have shown that not being employed full-time may have a negative influence on health,..’.

INTRODUCTION

3. Editor comment:

We advise that you abbreviate the introduction to keep its focus consistent with the aim of the study. Given that the study's objective is exploratory, it appears excessive space is allocated to specific factors, such as occupation. Please shorten the introduction accordingly and realign it with the study's aims.

Response from authors:

This manuscript is the outcome of a project supported by a research grant that was very specific in its objectives - to explore risk factors for developing LTCs and progression to multimorbidity in a working-age population. Employment status was specifically included within a wide range of exploratory variables with the additional objective for employment status to receive specific focus as a secondary aim. The current introduction reflects these objectives, and we cannot see how any further amendments would benefit this section. We would like to highlight the comment made by the first reviewer in relation to the usefulness of including employment data (“This is an interesting and well-written paper. Its strengths include the longitudinal data and the data on employment - which is particularly valuable in the context of multimorbidity among working-age people”). 

METHODS

4. Editor comment:

It is necessary to provide a clearer account of those individuals who did not participate in the second wave of the study and were subsequently excluded. We recommend including a flowchart as a supplementary file, outlining the participant trajectory. Moreover, we ask that you supply a table comparing the baseline characteristics of participants who met the inclusion criteria but did not participate in the follow-ups.

Response from authors:

The data set used for this secondary data analysis study only included participants who had completed both the SELCoH 1 and SELCoH 2 surveys and were within a working-age range. Analysis of the SELCoH data not included in our dataset was outside the aims and objectives of this study. However, there have been many previous SELCoH studies published with information on the original data, and the participant trajectory between SELCoH 1 and SELCoH 2 has been previously reported in studies using the complete SELCoH dataset (For example, Hatch et al., 2016; doi:10.1007/s00127-016-1191-x, and Harber-Aschan et al., 2019; doi: 10.1016/j.jpsychores.2018.11.005). Hatch et al., (2016) also has a supplementary table comparing baseline demographic characteristics of the SELCoH 1 and SELCoH 2 participants. We have now inserted the following sentence into the methods so that these papers are now referred to (the references have also been adjusted accordingly):

Methods page 6, lines 125-127:

Demographic characteristics of participants who responded at both waves, and reasons for non-response at the second wave have previously been reported [14, 15].

Please also see our previous response to the second reviewer in regard to participants who did not complete SELCoH 2, as we were able to establish that the 21 non-responders for health reasons or death (referred to in Harber-Aschan et al., 2019) were outside the age range of our study so would not have been included in any analysis of non-responders

5. Editor comment:

We noted that you group certain conditions together that are treated as a single condition in your analysis, as per Stagg et al. (2022). In this regard, please provide a rationale for the validity of classifying subjects as having a single condition when they may have multiple LTCs. Addressing this issue is crucial, as it may necessitate significant alterations in your analysis.

Response from authors:

This is a very good point, and we agree that this needs further clarification as no subject was classified as having one condition when they had multiple conditions, and this is not clearly stated in the methods section. During the lengthy LTC data coding process, careful attention was given to the number of individual self-reported conditions when developing the final groupings of conditions to ensure that an individual with two or more conditions always had these conditions listed separately, and not within a single condition group. Once the list of LTC groups were established, a further check was made within Stata by viewing all the condition groups against each participant ID to confirm that the number of conditions within each group for each participant was never more than 1. This process was essential to determining the correct number of LTCs self-reported by each participant. 

To ensure clarity on this point, we have amended the long-term conditions section of the methods as follows: 

Methods page 7, lines 140-144. This sentence has been amended so it now reads: ‘The final list of LTCs was constructed by allocating conditions either into specific condition groups (for example, two condition groups were asthma and high blood pressure), or into groups of similar conditions where individual case numbers were <5 (for example, gastrointestinal conditions and respiratory conditions).’

 Methods page 7, line 144-145: The following sentence was also inserted: ‘No participant had more than one condition within any of the individual combined condition groups.’ 

6. Editor comment:

Your exclusion of individuals whose health status improves may introduce significant selection bias. The current emphasis on studying individuals with a deteriorating health status does not sufficiently justify your choice of analysis. We strongly suggest revising your analysis to include all participants. 

Response from authors:

Whilst ‘improvers’ were identified in the data and was discussed in the research team as it is interesting, they were not included as an additional LTC trajectory group primarily as this group was never included in the study protocol, and we were confident that excluding it was not detrimental to the key aims of the study focusing on participants reporting increased numbers of LTCs (this was acknowledged in the methods section, page 9, lines 207-209). The ‘improvers ‘group was also complex and comprised sub-groups with some participants reporting 1 LTC at S1 and none at S2, and another group reporting more than 1 LTC at S1 and fewer LTCs at S2. There were additional potential limitations in analysing this group as we did not know whether they improved due to successful on-going treatments, or had experienced a complete recovery, or were in remission (this included cancer) which would have been a major limitation to the findings. 

From a statistical perspective, the multinomial regression used in the study comprises the two discrete categorical outcome groups (0 to ≥1 LTCs and 1-to-many LTCs), and inclusion of the improvers group would not have impacted the results of these two categorical outcomes, or imply an error in the results of these two categories due to selection bias. 

However, had this study focused more generally on all LTC trajectories (and included comparative patient medical records), a significantly larger sample size would have enabled an in-depth stratified analysis of the ‘improved’ groups of participants. We have acknowledged the omission of this group in the limitation section of the discussion, and have added in ‘and the addition of medical record data’:

Discussion page 23, lines 417-421: “ A larger sample size and the addition of medical record data would also enable factors supporting positive health transitions to be investigated. By expanding the scope of future studies to include positive health transitions, this would provide guidance for health promotion interventions aimed at slowing the journey to multimorbidity”. 

DISCUSSION

7. Editor comment:

Please refine the structure of your discussion for improved readability. For instance, the correlation between employment categories is discussed at two different points (lines 318 and 338). Consolidating this discussion into a single paragraph would greatly enhance readability. 

Response from authors:

The discussion has now been extensively revised, with the employment section consolidated into a single paragraph and various additional edits made to enhance readability through this section. Please see the tracked changes in the discussion pages 19-23. 

8. Editor comment:

Also, we recommend limiting speculative conclusions, as your study merely observes associations (e.g., employment can both be a potential causal factor for multimorbidity, or employment could reflect health status at baseline which is the real risk factor for multimorbidity).

Response from authors:

The revisions to the discussion have taken this point into consideration and made appropriate changes so that speculative conclusions are limited. Please see the changes made to the discussion listed in our response to editor comment 2, as these two comments are linked. 

The revisions to the discussion also specifically ensure that there is no suggestion of causality in relation to the regression results between employment and the LTC trajectories. We are aware that there may be a bi-directional association between employment and health and acknowledged this in the following sentence in the discussion:

Discussion page 21, lines 358-360: ‘One possibility is that there is a higher proportion of people with one LTC at baseline who found the need to change from full-time to part-time employment, and so this group may already be at increased risk.’ 

This has been further highlighted in the limitations section of the discussion with the insertion of the following sentence:

Discussion page 23 lines 424-426: ‘Also, reverse causality cannot be excluded, particularly in the case of employment where health status may lead to reduced working hours or unemployment’. 

9. Editor comment:

Also, please reconsider carefully your limitation statement. Consider for example the limited number of LTCs, the grouping of LTCs, the lack of information on propensity to seek care (and thus receive a diagnosis).

Response from authors:

The limitations section has been expanded and the following comment inserted into the discussion on page 23, lines 411-417. 

‘In contrast to many studies using patient records, all self-reported conditions were included in this study. However, a larger sample size would have enabled a larger number of LTCs to be analysed separately rather than being combined into similar condition groups. Additionally, despite the advantages of including all self-reported conditions, it should also be noted that without access to medical records, we were unable to confirm whether a formal diagnosis had been received for all conditions, or if there were un-reported conditions that had not yet received a diagnosis or were being successfully managed by the individual.’

---

## [Editor Report · Decision Letter 2]

29 Aug 2023

Risk factors for the progression to multimorbidity among UK urban working-age adults. A community cohort study

PONE-D-22-33742R2

Dear Dr. Stagg,

We’re pleased to inform you that your manuscript has been judged scientifically suitable for publication and will be formally accepted for publication once it meets all outstanding technical requirements.

Kind regards,

Andrea Dell'Isola

Academic Editor

PLOS ONE

---

## [Editor Report · Acceptance letter]

1 Sep 2023

PONE-D-22-33742R2 

Risk factors for the progression to multimorbidity among UK urban working-age adults. A community cohort study. 

Dear Dr. Stagg:

I'm pleased to inform you that your manuscript has been deemed suitable for publication in PLOS ONE. Congratulations! Your manuscript is now with our production department. 

Kind regards, 

on behalf of

Assoc Prof Andrea Dell'Isola 

Academic Editor

PLOS ONE